# Fabrication and Properties of the Multifunctional Rapid Wound Healing *Panax notoginseng*@Ag Electrospun Fiber Membrane

**DOI:** 10.3390/molecules28072972

**Published:** 2023-03-27

**Authors:** Zhaoju Gao, Songlin Liu, Shangfei Li, Xinzhe Shao, Pingping Zhang, Qingqiang Yao

**Affiliations:** NHC Key Laboratory of Biotechnology Drugs (Shandong Academy of Medical Sciences), Key Lab for Rare & Uncommon Diseases of Shandong Province, School of Pharmacy and Pharmaceutical Sciences & Institute of Materia Medica, Shandong First Medical University & Shandong Academy of Medical Sciences, Jinan 250117, China

**Keywords:** electrospun nanofiber membrane, *Panax notoginseng*, silver nanoparticles, core/shell, multifunction

## Abstract

The *Panax notoginseng*@Ag core/shell electrospun fiber membrane was prepared by coaxial electrospinning combined with the UV reduction method (254 nm). The prepared *Panax notoginseng*@Ag core/shell nanofiber membrane has a three-dimensional structure, and its swelling ratio could reach as high as 199.87%. Traditional Chinese medicine *Panax notoginseng* can reduce inflammation, and the silver nanoparticles have antibacterial effects, which synergistically promote rapid wound healing. The developed *Panax notoginseng*@Ag core/shell nanofiber membrane can effectively inhibit the growth of the Gram-negative bacteria *Escherichia coli* and the Gram-positive bacteria *Staphylococcus aureus*. The wound healing experiments in Sprague Dawley mice showed that the wound residual area rate of the *Panax notoginseng*@Ag core/shell electrospun nanofiber membrane group was only 1.52% on day 9, and the wound of this group basically healed on day 12, while the wound residual area rate of the gauze treatment group (control group) was 16.3% and 10.80% on day 9 and day 12, respectively. The wound of the *Panax notoginseng*@Ag core/shell electrospun nanofiber membrane group healed faster, which contributed to the application of the nanofiber as Chinese medicine rapid wound healing dressings.

## 1. Introduction

Wound healing is a complex pathophysiological process, with four phases: hemostasis, inflammation, proliferation, and remodeling [1,2]. In the inflammatory phase, large amounts of wound exudate are prone to occur at the wound [3,4]. However, excess exudate favors the growth and reproduction of bacteria, which can lead to an intensified inflammatory response, slow down the rate of wound healing, and make wound healing stop in the inflammatory phase [5,6,7]. Therefore, a suitable moist environment to effectively absorb the wound exudate and maintain wound healing has become a research hotspot [8,9,10,11]. Natural polymer chitosan (CS) has good water absorption and biocompatibility, and is widely used in wound dressings [12,13,14,15,16]. The invasion of external environment bacteria into wounds is also an important factor affecting wound healing [17,18]. Silver nanoparticles (Ag NPs) can play a bacteriostatic role by increasing the permeability of the bacterial membrane [19], interfering with DNA replication [20], and denaturing bacterial proteins [21].

*Panax nototoginseng* (PN) has been used in wound healing for hundreds of years [22,23,24]. The traditional Chinese medicine PN could reduce inflammation. PN could suppress retinoic acid-inducible gene I (RIG-I)-like receptors (RLRs), tumor necrosis factor-2 (TNF-2), nuclear factor-kappa B (NF-κB), tumor necrosis factor-α (TNF-α), and interleukin-8 (IL-8), which significantly inhibit the increase in capillary permeability, inflammatory exudation, tissue edema, leukocyte migration and granulation tissue proliferation caused by acute inflammation [25]. Saponins are the effective components of PN and have been widely studied for their good regulatory effects in wound healing. Saponins can promote angiogenesis in the wound by activating Notch/VEGFR-2 or Ang2/Tie2 signaling pathways [26,27]. Studies showed that saponins can inhibit the scar formation of the wound by inhibiting the proliferation of fibroblasts and reducing the expression of α-smooth muscle actin (α-SMA) in skin wounds. Currently, the clinical dosage forms of PN in wound healing mainly include PN powder and PN capsules. PN powder is directly applied to the wound and is often fixed with gauze. The high permeability of gauze makes it easy to dehydrate the wound, and gauze can easily adhere to the wound; the wound is painful and bleeding when replacing gauze, which causes mechanical damage, and the number of dressing changes is frequent, time-consuming and laborious; patients find the process painful, and it is easy for bacteria to invade after the gauze is soaked, resulting in a high chance of wound infection. In addition, PN powder has a high local drug concentration when it is externally used, which can stimulate the skin and easily cause a skin allergic reaction. PN capsule oral preparation has the limitation of a fast elimination rate. In addition, patients need frequent administration to maintain an effective blood concentration, resulting in poor patient compliance.

Electrospinning is a strategy that can be used to prepare nanofiber membranes by electrostatic force [28]. Coaxial electrospinning is an electrospinning technique in which the nuclear layer spinning solution and shell spinning solution are placed into the nuclear layer syringe and shell syringe, respectively, and the core/shell nanofibers are prepared by a concentric needle device [29,30,31,32]. The composition, structure, and size of the coaxial electrospinning membrane can be easily adjusted. Traditional Chinese medicine and active substances can be added to the spinning system according to needs, and the self-supporting nanofiber membrane can be constructed by adjusting the spinning condition and process to regulate its structure and size, and it can be applied directly to the wound to promote wound healing. The three-dimensional structure of the coaxial electrospun nanofiber membrane is similar to that of the extracellular matrix (ECM), which is conducive to gas exchange at the wound and the adhesion of fibroblasts, thus accelerating the formation of granulation tissue in the wound [33,34,35,36].

Because of the antibacterial effect of silver, there are many studies on silver nanoparticle-loaded electrospun fibers for wound healing, but there is no study on electrospun fibers that combine antibacterial and anti-inflammatory effects to promote wound healing. The use of traditional Chinese medicine to inflammatory conditions is an attractive approach to promote wound healing, and some natural active ingredient-loaded electrospun fiber membranes have been developed. However, electrospun fibers loaded with PN have not been reported. In this study, CS was selected as the water-absorbing active component, silver nitrate as the silver source, and Chinese medicine PN as the medicine. The mixed solution of PN, CS, and polycaprolactone (PCL) was used as the core electrospinning solution, and the mixed solution of silver nitrate and polyvinylpyrrolidone (PVP) was used as the shell electrospinning solution for coaxial electrospinning. Combined with UV reduction, *Panax notoginseng*@Ag (PN@Ag) core/shell nanofiber membranes loaded with the Chinese medicine PN and the active substance Ag NPs were prepared. The addition of CS enhances the hygroscopic property of PN@Ag core/shell electrospun fiber membranes. The CS in the PN@Ag electrospun fiber membrane provides a suitable moist environment for the wound, thus speeding up wound healing to a certain extent. PN reduces inflammation, while the Ag NPs effectively inhibit the growth of bacteria in the wound site, thus synergistically promoting rapid wound healing. Antibacterial experiments showed that the PN@Ag electrospun fiber membrane can effectively inhibit the growth of *Escherichia coli* and *Staphylococcus aureus*. In vivo experiments of Sprague Dawley mice indicated that the PN@Ag electrospun fiber membrane group had the best wound healing compared with the gauze treatment group (control group). The constructed PN@Ag core/shell nanofiber membrane provided a new and effective strategy for rapid wound healing.

## 2. Results and Discussion

### 2.1. Characterization of the PN@8 wt%Ag-2 h Core/Shell Fiber Membrane

Figure 1 shows the XRD patterns of the PN powder and PN@8 wt%Ag-2 h nanofiber membrane. The PN powder had sharp characteristic diffraction peaks at 21.48° and 23.82°, while the PN@8 wt%Ag-2 h nanofiber membrane had a weak diffraction peak at 21.48°, indicating that the crystallinity of the PN powder in the nanofiber decreased, which is beneficial to the sustained release of the PN powder from nanofibers [37]. Appendix A shows the SEM of PN used for electrospinning. Figure 2a,b are the SEM of the PN@8 wt%Ag-2 h nanofiber membrane. It can be seen from Figure 2a,b that the fibers in the prepared nanofiber membrane are continuous and the surface is smooth. PN@6 wt%Ag-2 h and PN@10 wt%Ag-2 h have a similar morphology (Appendix A). The fibers in the membrane are randomly stacked and constructed into a three-dimensional structure similar to ECM, which provides structural support and a survival microenvironment for tissue cells and speeds up wound healing. The distribution of fiber diameters in the fiber membrane was uniform. The diameter distribution of the PN@8 wt%Ag-2 h electrospun fiber was measured by Image J software (2.1.0), and the average diameter was 100 ± 15 nm.

Figure 3a–c are TEM images of fiber membranes of PN@6 wt%Ag-2 h, PN@8 wt%Ag-2 h, and PN@10 wt%Ag-2 h, respectively. It can be seen from the figures that silver nitrate is reduced to form Ag NPs and dispersed in the electrospun nanofiber. The XPS spectra also indicated the formation of Ag NPs in the electrospun nanofiber membrane (Appendix A). The morphology of Ag NPs has great influence on the bacteriostatic effect of the electrospun nanofiber membrane. The smaller the size of Ag NPs, the better the bacterial inhibition effect of the nanofiber membrane. Furthermore, the more uniform the distribution of Ag NPs in the nanofiber membrane, the better the bacterial inhibition effect of the nanofiber membrane [38,39,40]. It can be seen from Figure 3 that the amount of silver nitrate affects the particle size and distribution form of Ag NPs in the electrospun nanofiber. The particle size of Ag NPs in the electrospun nanofiber was measured by Image J software (2.1.0), and the particle size distribution was statistically analyzed.

The size of Ag NPs in the core/shell electrospun nanofiber membranes of PN@6 wt%Ag-2 h, PN@8 wt%Ag-2 h, and PN@10 wt%Ag-2 h was 3.76 ± 0.04 nm (Figure 3d), 3.99 ± 0.18 nm (Figure 3e), and 4.86 ± 0.12 nm (Figure 3f), respectively. In PN@6 wt%Ag-2 h, the Ag NPs smaller than 6 nm accounted for more than 95%, and a few Ag NPs larger than 10 nm were generated, which may have been caused by Ag NPs agglomeration [41]. The Ag NPs size distribution in PN@8 wt%Ag-2 h is uniform (Figure 3e), and all Ag NPs are in the range of 2–6 nm. In PN@10 wt%Ag-2 h, Ag NPs smaller than 6 nm accounted for approximately 60%, and a large proportion of Ag NPs with a large size appeared in the fiber, and some Ag NPs even reached 16 nm (Figure 3d).

### 2.2. The Swelling Ratio of Prepared Nanofiber Membranes

Electrospun fiber membranes have high porosity and good air permeability, and can efficiently absorb tissue fluid at the wound [42,43,44]. Comparing Table 1 and Table 2, the average swelling ratio of the PN@8 wt%Ag-2 h fiber membrane is significantly higher than that of the PN@8 wt%Ag-NCS fiber membrane, which shows that the addition of CS enhances the hygroscopic property of electrospun fiber membranes. This is because the molecular structure of CS contains hydrophilic groups such as amino and hydroxyl groups, which enhance the hygroscopic properties of the fiber membrane [45,46]. Studies have shown that the high swelling ratio of the PN@8 wt%Ag-2 h electrospun fiber membrane could provide a suitable moist environment for the wound, thus speeding up the wound healing [47,48].

### 2.3. Antibacterial Results of the PN@Ag Electrospun Nanofiber Membrane

The fiber membranes (PN@8 wt%Ag-2 h and PN@6 wt%Ag-2 h) with small particle size and uniform distribution of Ag NPs in fibers were selected to study their antibacterial properties, as shown in Figure 4 and Figure 5. It can be seen from the figure that there is no bacteriostatic zone in the control group PCL@PN (Figure 4a and Figure 5a), indicating that PCL@PN does not have any bacteriostatic effect on *Escherichia coli* and *Staphylococcus aureus*, while PN@8 wt%Ag-2 h (Figure 4c and Figure 5c) and PN@6 wt%Ag-2 h (Figure 4b and Figure 5b) have more obvious bacteriostatic zones and better antibacterial effects. The diameter of the bacteriostatic zone in the experimental group was measured by Image J software (2.1.0). The diameter of the bacteriostatic zone in the PN@8 wt%Ag-2 h group was approximately 2.23 cm (Figure 4c), and that in the PN@6 wt%Ag-2 h group was approximately 2.19 cm (Figure 4b), indicating that the inhibitory effect of PN@8 wt%Ag-2 h on *Escherichia coli* was better than that of PN@6 wt%Ag-2 h. The diameter of the bacteriostatic zone in the PN@8 wt%Ag-2 h group was approximately 2.11 cm (Figure 5c), and that in the PN@6 wt%Ag-2 h group was approximately 1.96 cm (Figure 5b), indicating that the inhibitory effect of PN@8 wt%Ag-2 h on *Staphylococcus aureus* was better than that of PN@6 wt%Ag-2 h. The inhibitory effect of PN@8 wt%Ag-2 h on *Escherichia coli* and *Staphylococcus aureus* was better than that of PN@6 wt%Ag-2 h, which could be attributed to the higher Ag NP content and smaller particle size in PN@8 wt%Ag-2 h.

### 2.4. Cytotoxicity of Electrospun Nanofiber Membranes

The in vitro toxicity of nanofiber membranes was determined by the CCK-8 method. The leachate of five different concentration gradients of PN@8 wt%Ag-2 h and blank membranes acted on fibroblasts (L929 cells). Even at higher concentrations (100 μg/mL), PN@8 wt%Ag-2 h and blank membranes showed no toxicity (the cell viability was more than 80%). The results showed that the fiber membrane carrier and drug-loaded PN@8 wt%Ag-2 h were safe and non-toxic, and did not affect the cell viability during the proliferation stage of wound healing (Figure 6).

### 2.5. In Vivo Wound Healing in Sprague Dawley Mice

The wound condition was photographed on days 0, 3, 6, 9 and 12 to record the wound healing in Sprague Dawley mice, as shown in Figure 7. On day 3 of treatment, whitening of the wound was observed in the control group and blank membrane treatment group, which indicated that there were symptoms of infection and inflammation in the wound, and the wound inflammation was more serious in the control group (gauze treatment). However, no significant signs of inflammation were seen in the PN@8 wt%Ag-2 h treatment group, the PCL@PN treatment group, or the PVP@Ag treatment group.

Wound healing was characterized by the percentage of wound area treated with fiber membranes to the initial wound area (wound aera (%)) (Figure 8). In the PN@8 wt%Ag-2 h group, the wound area was 52.94% on day 3, 16.07% on day 6, 1.52% on day 9, and 0.46% on day 12. On the day 12, the wound area (%) of the PCL@PN, PVP@Ag, blank membrane and control was 2.24%, 4.07%, 7.69% and 10.80%, respectively. The results showed that the PN@8 wt%Ag-2 h electrospun core/shell fiber membrane had a significantly higher wound closure rate than the other four groups and the wound in the PN@8 wt%Ag-2 h electrospun core/shell fiber membrane group was smooth. In addition, there was no obvious scab formation in the PN@8 wt%Ag-2 h group compared with the gauze treatment group (control group). The good rapid healing effect of the wound was attributed to the synergy role of Ag NPs and the gradually released of PN in the prepared nanofiber (Appendix A).

## 3. Experimental Section

### 3.1. Materials

Polycaprolactone (PCL, M_W_ = 8 × 10^4^), polyvinylpyrrolidone K-90 (PVP, M_W_ = 1.3 × 10^6^), chitosan, and ammonium bicarbonate were provided by Dalian Meilun Biological Co., Ltd. (Dalian, China). Chloroform (CHCl_3_) was provided by Tianjin Fuyu Fine Chemical Co., Ltd. (Tianjin, China). *Panax notoginseng* powder was obtained from Yunnan Notoginseng Science and Technology Pharmaceutical Co., Ltd. (Yunnan, China). Sinopharm Chemical Reagent Co., Ltd. (Shanghai, China) provided the silver nitrate (AgNO_3_), ethanol (C_2_H_5_OH), and N,N-dimethylformamide (DMF). *Escherichia coli* was purchased from Beijing Beina Chuanglian Institute of Biotechnology (Beijing, China), and *Staphylococcus aureus* was provided by Shanghai Bioresource Collection Center (Shanghai, China).

### 3.2. Preparation of Electrospun Fiber Membranes

In a typical experiment, PCL (0.8 g), CS (0.12 g), and ammonium bicarbonate (0.024 g) were dissolved in CHCl_3_/DMF (*v*/*v*, 8 mL:2 mL). After stirring for 2 h at 25 °C until completely dissolved, 0.08 g of PN powder (accounting for 10 wt% of the PCL mass) was added, and then stirred for 1 h at 25 °C to obtain core spinning solution. PVP (1.5 g) was dissolved in a solution of C_2_H_5_OH/DMF (*v*/*v*, 7 mL:3 mL). After being dissolved at 25 °C, 0.12 g of AgNO_3_ (accounting for 8 wt% of the PVP) was added. Then, the shell electrospinning solution was obtained by stirring in the dark. The core and shell electrospinning solution were coaxially electrospun under the following electrospinning conditions: 25 °C temperature, 30% humidity, the positive and negative voltages applied to the syringe tip were 21 and 0.15 kV, respectively, the distance from the tip of the syringe needle to the collector is kept at a constant 18 cm, the flow rate of the core spinning solution was 0.2 mm/min, and the speed of the shell spinning solution was 0.4 mm/min. The wet fiber membrane was obtained by coaxial electrospinning, then the PN@8 wt%Ag core/shell electrospun nanofiber membrane was obtained after drying at 50 °C for 3 h. The silver nitrate in the electrospun nanofiber membrane was reduced to Ag NPs after irradiation with 254 nm UV light for 2 h, and PN@8 wt%Ag-2 h was obtained. The concentration of the marker of PN (ginsenoside Rg1) in PN@8 wt%Ag-2 h was 1.45‰ (the weight of the ginsenoside Rg1/the weight of the finished nanofiber membrane).

The PCL@PN fiber membrane loaded with PN was obtained by uniaxial electrospinning of the core spinning solution. After the shell spinning solution was uniaxially electrospun and irradiated with 254 nm UV light for 2 h, the PVP@Ag fiber membrane loaded with Ag NPs was obtained. The core spinning solution remained unchanged, and the amount of AgNO_3_ in the shell spinning solution was changed. 0.09 g AgNO_3_ (accounting for 6 wt% of the PVP mass) and 0.15 g AgNO_3_ (accounting for 10 wt% of the PVP mass) were added to the shell spinning solution, and the core/shell fiber membranes PN@6 wt% Ag and PN@10 wt%Ag were obtained under the same electrospinning conditions. PN@6 wt%Ag-2 h and PN@10 wt%Ag-2 h were obtained after 254 nm UV irradiation for 2 h. Under the same conditions (irradiation for 2 h), the PN@8 wt%Ag-NCS core/shell fiber membrane without CS was prepared with the same shell spinning solution and no CS in the core spinning solution. Blank core/shell fiber membranes were prepared under the same conditions without AgNO_3_ in the shell spinning solution or PN in the core spinning solution. The corresponding name of each prepared sample is shown in Table 3.

### 3.3. Characterizations

The crystal phase of the PN@8 wt%Ag-2 h nanofiber membrane was detected by an X-ray diffraction (XRD) spectrometer (Rigaku D/Max 2200PC diffractometer) with a scanning range from 10° to 80°. The morphology and the Ag NPs loading status of the electrospun nanofibrous membranes were uncovered by a field emission scanning electron microscope (FE-SEM, JSM-6700F) and a transmission electron microscope (TEM, JEM-1011). The Image J software (2.1.0) was used to analyze the fiber diameter in the electrospun nanofiber membrane and the size of Ag NPs in the nanofiber membrane. The water absorption of PN@8 wt%Ag-2 h was determined by the swelling index [49]. PN@8 wt%Ag-2 h was cut into squares (3 cm × 3 cm), weighed and recorded as M_0_. The PN@8 wt%Ag-2 h square fiber membrane was placed in phosphate buffer solution (pH = 7.4) and swelled at 37 °C for 30 min, then one end of the sample was clamped with tweezers for 30 s until there were no more water droplets. The excess liquid on the surface of the fiber membrane was removed with filter paper, and the weight of the fiber membrane was recorded as M_1_. The experiments were performed on three samples and the swelling ratio was calculated using Equation (1). The swelling ratio of the PN@8 wt%Ag-NCS fiber membrane was tested by the same method for comparison.
(1)$$\mathrm{Degree}\;\mathrm{of}\;\mathrm{swelling}=\frac{M_{1}-M_{0}}{M_{0}}\times 100\%$$
where M_1_ is the weight of the fiber film before drying, and M_0_ is the weight of the fiber film after drying.

### 3.4. Antibacterial Evaluation Assay

The inhibition effect of the nanofiber membrane on bacteria (*Escherichia coli* and *Staphylococcus aureus*) was studied by an antibacterial ring test [49]. First, we mixed the bacteria powder of *Escherichia coli* or *Staphylococcus aureus* with sterile water, and then poured it all into a conical flask filled with beef extract peptone liquid culture medium. The conical flask was then placed in a shaker (37 °C, 220 rpm), and the original bacterial solution was obtained after 12 h. Subsequently, the concentration of the bacterial solution was diluted to 1 × 10^8^ CFU/mL with fresh sterilized beef extract peptone liquid medium. The diluted *Escherichia coli* or *Staphylococcus aureus* was evenly smeared on culture plates, and then PCL@PN, PN@6 wt%Ag-2 h, and PN@8 wt%Ag-2 h fiber membrane discs (1.5 cm × 1.5 cm) were applied in the center of the culture plates and cultured at bacteriological incubator (37 °C, 24 h). The antibacterial evaluation assay of the fiber membrane on *Escherichia coli* and *Staphylococcus aureus* was determined by the diameter of the bacteriostatic zones.

### 3.5. Cytotoxicity of Electrospun Nanofiber Membrane

Fibroblasts (L929 cells) (Procell Life Science&Technology Co., Ltd., Wuhan, China) were cultured in MEM medium at 5% CO_2_, 37 °C. The cytotoxicity of PN@8 wt%Ag-2 h was measured via the Cell Counting Kit-8 (CCK-8) assay. To summarize, L929 cells digested by trypsin were seeded in 96-well plates at a concentration of 5 × 10^5^cells per well. After incubating cells in the cell incubator for 24 h, MEM medium extract supplemented with two different fiber membranes was used as the experimental group. The mass fraction (the mass fraction of the fiber membrane mass in the MEM medium) of the fiber membrane in the PN@8 wt%Ag-2 h experimental group and blank membrane experimental group was set according to the gradient of 5, 10, 20, 50, and 100 μg/mL. A well with 100 μL MEM medium was used as the control group. Following incubation 24 h, CCK-8 was added and incubated for another 2 h. The optical density (OD) value at 450 nm was recorded using a microplate reader (Berthold TriStar^2^S LB 942). The OD value of the control group corresponded to 100% cell activity and the viability was calculated as Equation (2). Each group was used with three wells.
(2)$$\;\;\mathrm{Cell}\;\mathrm{Viability}\;=\frac{{\mathrm{OD}}_{\mathrm{drug}\;}-{\mathrm{OD}}_{\mathrm{PBS}}}{{\mathrm{OD}}_{\mathrm{control}\;}-{\mathrm{OD}}_{\mathrm{PBS}}}$$
where OD_drug_ was the OD value of PN@8 wt%Ag-2 h or blank membrane, OD_PBS_ was the OD value of the PBS solution, and OD_control_ was the OD value of the control group.

The pairwise comparisons for groups were performed using a one-sample *t*-test. *p* < 0.05 was considered statistically significant.

### 3.6. Establishment of Wounds in Mice and Wound Healing In Vivo

The wound healing experiment was conducted with male Sprague Dawley mice weighing 160–200 g. The animal study protocol was approved by Principles of Laboratory Animal Care (People’s Re-public of China) and the Institutional Review Board of Shandong First Medical University, China. (protocol code W202210280253 and date of approval is 28 October 2022). The mice were anaesthetized with 3% pentobarbital sodium and depilated, and a circular full skin wound model (the diameter of the wound was 10 mm) was constructed in their abdomen with a sterilized punch. To avoid individual differences, the mice were randomly divided into the PN@8 wt%Ag-2 h treatment group, PCL@PN treatment group, PVP@Ag treatment group, blank membrane treatment group, and control group (gauze treatment) (*n* = 3). The fibrous membrane with a diameter of 10 mm was disinfected and placed over the wound. The wound healing was recorded on day 0 (the skin wound model was established) and on day 3, 6, 9, and 12 after fiber membrane treatment. The wound healing was expressed as the percentage of the wound area after membrane treatment to the wound area on the 0th day, and the calculation formula is shown in Equation (3).
(3)$$\mathrm{Wound}\;\mathrm{area}(\%)=\frac{A}{A_{i}^{}}\times 100\%$$
where A_i_ is the wound area of the establishment of the skin wound model and A is the wound area after a fixed time.

## 4. Conclusions

In this paper, PN@Ag core/shell electrospun fiber membranes were prepared by electrospinning technology combined with the UV reduction method. A high swelling ratio (199.87%) of the PN@Ag core/shell electrospun nanofiber membrane could maintain a moist environment for wound healing. In the PN@Ag core/shell electrospun nanofiber, the traditional Chinese medicine PN and Ag NPs were co-loaded in the nanofiber membrane. PN could reduce inflammation, and the Ag NPs have antibacterial effects. In vitro experiments showed that the prepared nanofiber membranes can significantly inhibit the growth of *Escherichia coli* and *Staphylococcus aureus*. The synergistic effect of PN and Ag NPs proved the rapid wound healing role of PN@Ag core/shell electrospun nanofiber membranes. In vivo experiments showed that the wound residual area rate of the PN@Ag core/shell electrospun nanofiber membrane group was only 1.52% on day 9, and the wound of this group basically healed on day 12, while the wound residual area rate of the gauze treatment group (control group) was 16.3% and 10.80% on day 9 and day 12, respectively. In addition, the PN@Ag electrospun nanofiber membrane has no cytotoxicity, showing a potential application prospect in rapid wound healing. The fabrication of PN@Ag core/shell electrospun nanofiber membranes provides theoretical support for the development of new traditional Chinese medicine rapid wound healing dressings.

## Figures and Tables

**Figure 1 molecules-28-02972-f001:**
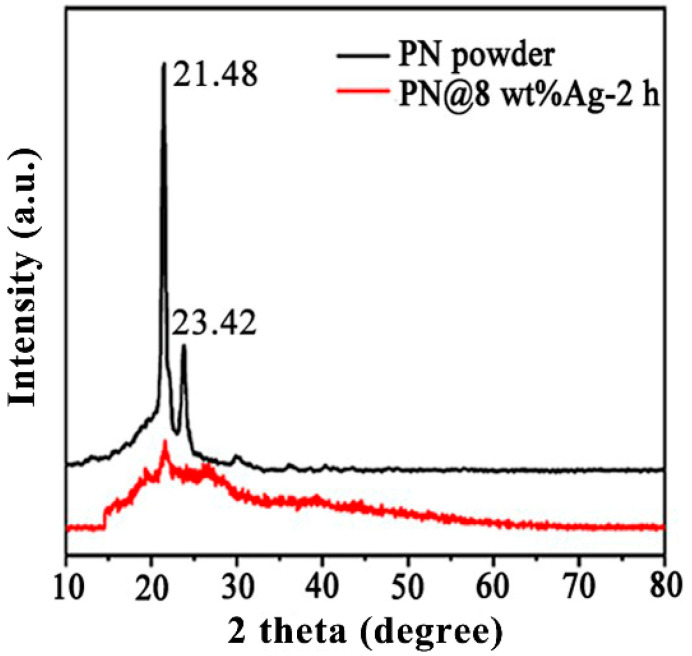
XRD patterns of PN powders, and the prepared PN@8 wt%Ag-2 h nanofiber membrane.

**Figure 2 molecules-28-02972-f002:**
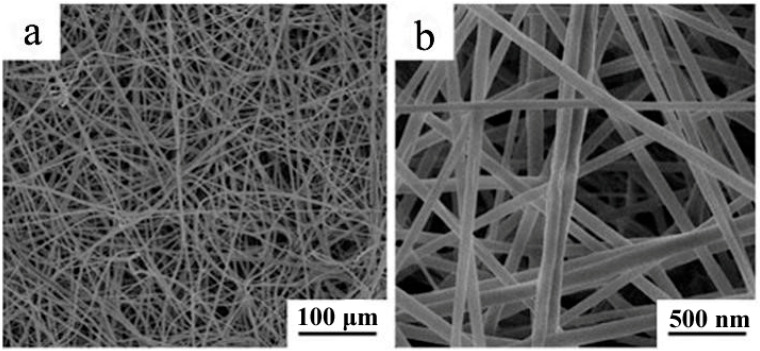
SEM images of the PN@8 wt%Ag-2 h nanofiber membrane (**a**,**b**).

**Figure 3 molecules-28-02972-f003:**
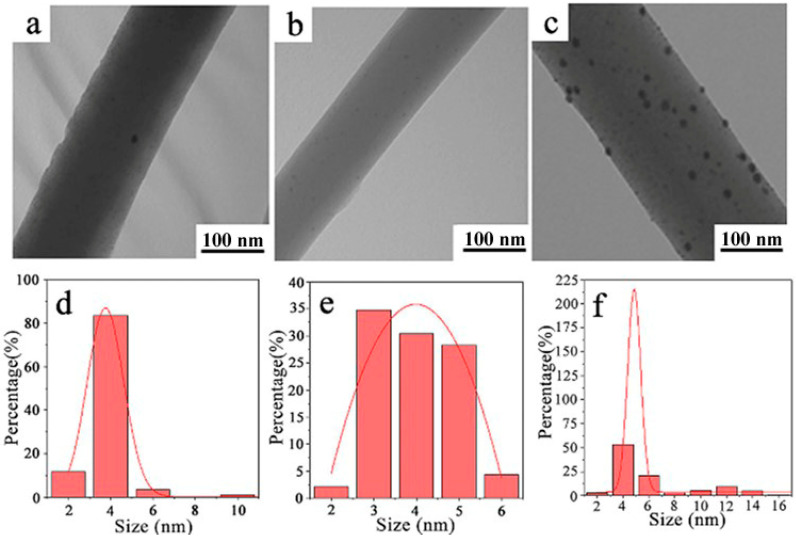
TEM images of PN@6 wt%Ag-2 h (**a**), PN@8 wt%Ag-2 h (**b**), PN@10 wt%Ag-2 h (**c**), and the particle size of Ag NPs and its distribution in PN@6 wt%Ag-2 h (**d**), PN@8 wt%Ag-2 h (**e**), and PN@10 wt%Ag-2 h (**f**).

**Figure 4 molecules-28-02972-f004:**
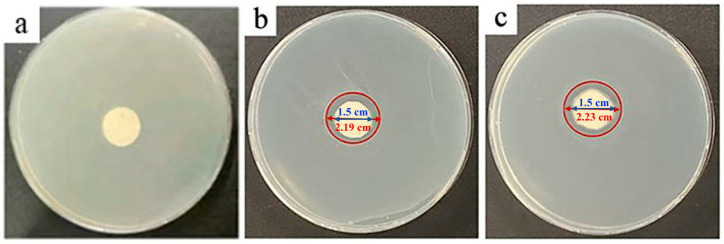
The inhibition zone of nanofibers (PCL@PN (**a**), PN@6 wt%Ag-2 h (**b**), and PN@8 wt%Ag-2 h (**c**)) on *Escherichia coli* after 24 h.

**Figure 5 molecules-28-02972-f005:**
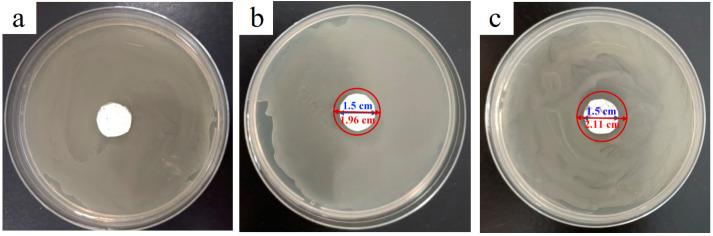
The inhibition zone of nanofibers (PCL@PN (**a**), PN@6 wt%Ag-2 h (**b**), and PN@8 wt%Ag-2 h (**c**)) on *Staphylococcus aureus* after 24 h.

**Figure 6 molecules-28-02972-f006:**
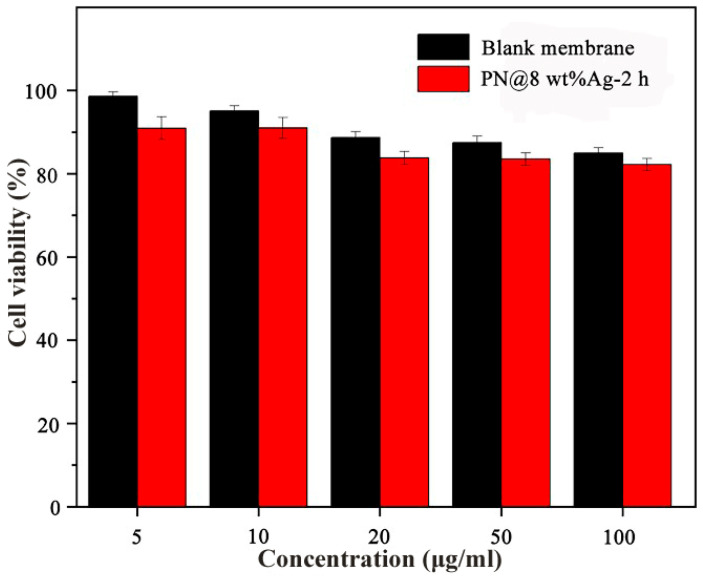
The viabilities of L929 cells after 24 h treatment with the PN@8 wt%Ag-2 h nanofiber membranes.

**Figure 7 molecules-28-02972-f007:**
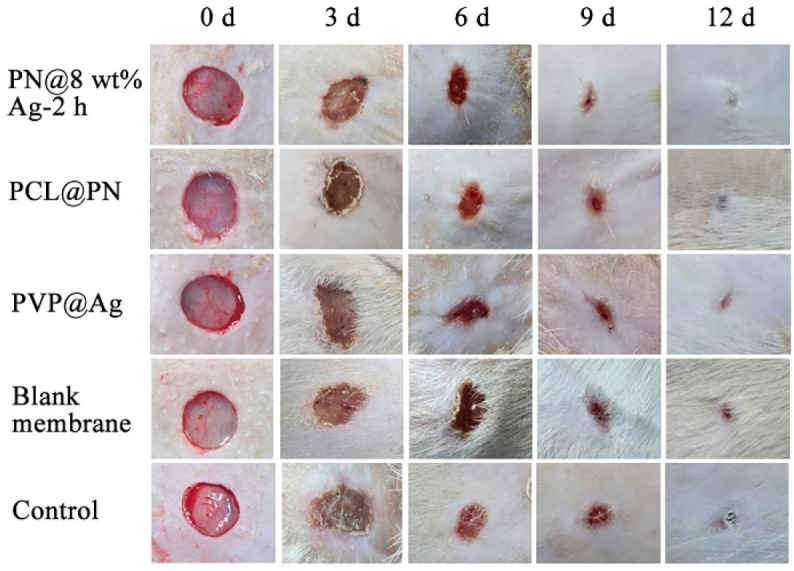
The effects of skin wound healing of Sprague Dawley mice in different treatment groups (PN@8 wt%Ag-2 h group, PCL@PN group, PVP@Ag group, blank membrane group, and control group) at day 0, 3, 6, 9 and 12 after operation.

**Figure 8 molecules-28-02972-f008:**
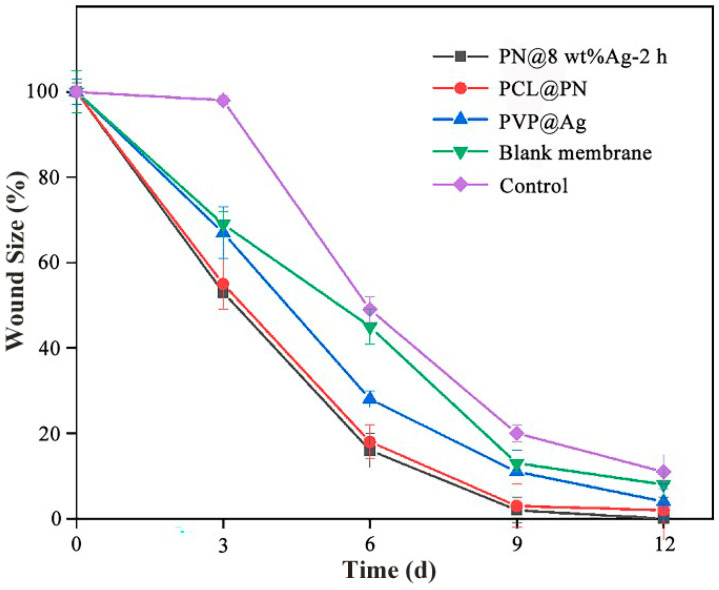
Wound area (%) of five treatment groups (PN@8 wt%Ag-2 h group, PCL@PN group, PVP@Ag group, blank membrane group, and control group) within 12 days. The data given are averages of the three mice in each group.

**Table 1 molecules-28-02972-t001:** Swelling ratios of PN@8 wt%Ag-NCS (control group).

Weight before Moisture Absorption M_0_ (g)	Weight After Moisture Absorption M_1_ (g)	Difference Value ΔM (g)	Swelling Ratio (%)	Average Swelling Ratio (%)
0.029	0.045	0.016	55.17	57.74
0.030	0.048	0.018	60.00	
0.031	0.049	0.018	58.06	

**Table 2 molecules-28-02972-t002:** Swelling ratios of PN@8 wt%Ag-2 h (experimental group).

Weight before Moisture Absorption M_0_ (g)	Weight after Moisture Absorption M_1_ (g)	Difference Value ΔM (g)	Swelling Ratio (%)	Average Swelling Ratio (%)
0.029	0.087	0.058	200.0	199.87
0.030	0.087	0.057	190.0	
0.031	0.096	0.065	209.6	

**Table 3 molecules-28-02972-t003:** The corresponding name of each sample.

PN in the Electrospun Core Solution (g)	AgNO_3_ in the Electrospun Shell Solution (g)	The Mass of CS in the Electrospun Core Solution (g)	The Irradiation Time of 254 nm Ultraviolet (h)	Sample Name
0.08	0.12	0.12	2	PN@8 wt%Ag-2 h
0.08	0	0.12	0	PCL@PN
0	0.12	0.12	2	PVP@Ag
0.08	0.09	0.12	2	PN@6 wt%Ag-2 h
0.08	0.15	0.12	2	PN@10 wt%Ag-2 h
0.08	0.12	0	2	PN@8 wt%Ag-NCS
0	0	0.12	0	Blank membrane

## Data Availability

Not applicable.

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
