# Peer review of "Fabrication and Properties of the Multifunctional Rapid Wound Healing Panax notoginseng@Ag Electrospun Fiber Membrane"

_molecules, 2023, doi:10.3390/molecules28072972_

Round 1

Reviewer 1 Report

General Comments:

In this report, Gao and Liu et al studied a Ag-doped hybrid nanofiber membrane (Panax notoginseng@Ag core/shell) for the purpose of anti-inflammation for would healing process. They utilized the coaxial electrospinning technology combined with the UV reduction method. The prepared Panax notoginseng@Ag core/shell nanofiber membrane has the three-dimensional structure, and good swelling ratio for adsorption. They adopted one traditional Chinese medicine Panax notoginseng to reduce inflammation, utilized silver nanoparticles for purpose of antibacterial effects, synergistically promoting rapid wound healing. The developed Panax notoginseng@Ag core/shell nanofiber membrane showed effective activity against the bacteria from in vivo experiments. Their studies are very interesting and of broad interests for materials, devices design, biomedical applications. However, the authors still needed to provide a more solid analysis and characterizations of the experiments. Moreover, the doping of Ag nanoparticles can indeed increase anti-bacteria effects and reduce infections but Ag NPs might also be toxic and should be concerned to the recovery of wound and normal cells. After all the necessary revisions, I recommend their manuscript to be considered for publication in Molecules.

1.     In abstract, “In vivo experiments in Sprague-Dawley mice showed that the wound residual area rate of the Panax notoginseng@Ag group was only 1.52% on day 9, and the wound of the Panax notoginseng@Ag group basically healed on day 12.” It should be more clear-cut if the authors put a rational comparison with the control group. (Like how fast the wound recovery with the fiber compared with regular ones)  

2.     The author stated, “The constructed PN@Ag core/shell nanofiber membrane has broad application prospects in the field of rapid wound healing dressing” Is there any requirements or standards for application of this hybrid fibers? Like types of wounds and depth of cut?

3.     The authors utilized silver nitrate (AgNO3) doped with PVP and followed UV reduction to obtain Ag atoms. Is there a way to find out how many Ag+ ions have been reduced to Ag? Characterizations of elements analysis like XPS might be used for study of the surface charges of Ag in this hybrid membranes.

4.     It is understandable that the using of Ag nanoparticles can increase anti-bacteria and reduce infections, but Ag NPs might also be toxic to the wound and normal cells. Analysis of potential biocompatibilities, doping amount of Ag should be discussed.

5.     What is the minimum doping amount (percentage) of Ag-NPs in fibers to achieve the desired anti-bacteria effects? It seems the good cell-viability of normal cells (L929) and anti-bacteria effects is conflict. How to reconcile that two parts?

6.     In the “Conclusion” part, the traditional Chinese medicine and Ag-NPs both contribute to the wound recovery, but from Figure 7, it seems the doping of Ag+ did not significantly change the wound recovery. Which one plays a more important role here?

Reviewer 2 Report

Manuscript is plagirised   upto 43% which is very high

No statistical model is mentioned 

ethical certificate for animal trial is missing

conclusion should be clear 

Reviewer 3 Report

The manuscript entitled “Fabrication and properties of the multifunctional rapid wound healing Panax notoginseng@Ag electrospun fiber membrane” shows interesting results for the development of the wound healing materials.  However, there are some comments for this manuscript as follows:

1. The authors should show the results of identification and characterization of Panax  notoginseng (PN) that was used as an active ingredient in the formulation of the electrospun fiber membrane.

2. The authors should specify the concentration of the marker (chemical/biological marker) of PN in the finished electrospun fiber membrane.  For example, % w/w and  the weight of the marker/the weight of the finished fiber membrane.

3. The authors should show the release profile of PN from the finished electrospun fiber membrane.

4. The authors should determine the anti-bacterial activities of the electrospun fiber membrane towards the gram-positive bacteria, e.g., Staphylococcus aureus, which is a common bacteria causing wound infection.

5. The authors should specify the source of L929 cells used in this study in the manuscript.

6. Apart from the cytotoxicity test in the L929 cells, the authors should determine the toxicity of the obtained electrospun fiber membrane in the skin fibroblast.

7. The authors should discuss on the mechanism of wound healing activities of the obtained electrospun fiber membrane.

8. Please consider to revise the last paragraph of the Introduction section.  It should not inform the results of the study in this section, however, it should state the research gap and the research objective instead.

Reviewer 4 Report

Fabrication and properties of the multi-functional rapid wound healing Panax notoginseng @Ag electrospun fiber membrane

The authors in this manuscript titled “Fabrication and properties of the multi-functional rapid wound healing Panax notoginseng @Ag electrospun fiber membrane” have conducted studies on using Panax notoginseng @Ag electrospun fiber membrane for wound dressing. The experiments were reported to be conducted in accordance with the Principles of Laboratory Animal Care (Peopl’e Republic of China) and the guidelines of the Animal Investigation committee Biology Institute of Shandong Academy of Science, China. Male Sprague – Dawley rats were obtained from an Pengyue Experimental Animal breeding company , Jinan China. The tests were carefully conducted and reported. The application of Chinese wound dressing and healing was demonstrated.

In the submitted paper the following queries are raised for the authors to revise:

Major Points

1. In Section 3.1 Figure 3, the particle sizes of Ag NPs and its distribution in PN@6wt% Ag - 2h and PN@10wt%Ag – 2h are not consistent as the other one. Why?

2. In the photographs shown after the 12th day for the change of size in wounds, except for PN@8wt% Ag - 2h, the others are quite similar, especially blank membrane and PVP@Ag. What does this infer?

3. Also for the control group between days 6 and 9 there could be seen a drastic reduction in wound area as compared to the other samples. Does this mean that natural wound healing effect starts working after 3 days?

Minor points

1. The XRD diffraction pattern misses Y-axis (Intensity a.u).

2. SEM images of only PN@8wt% Ag-2h is shown and others are not shown and XRD pattern of blank membrane and PN@8wt%-2h. Is there any reason for this?

  The paper may be accepted in the Journal Molecules as per the journal’s policy after minor revision and response for the queries raised.

Round 2

Reviewer 2 Report

Accept in present revised  form

Reviewer 3 Report

The authors corrected the manuscript following the reviewers’ suggestions, it thus could be accepted for publication.